# Characterization of Autochthonous Strains from the Cecal Content of Creole Roosters for a Potential Use as Probiotics

**DOI:** 10.3390/ani13030455

**Published:** 2023-01-28

**Authors:** Elvia Guadalupe Melara, Mavir Carolina Avellaneda, Ana Julia Rondón, Marlen Rodríguez, Manuel Valdivié, Yordan Martínez

**Affiliations:** 1Sustainable Tropical Agriculture, Zamorano University, San Antonio de Oriente, Francisco Morazán, Tegucigalpa 11101, Honduras; 2Plant Pathology, Diagnosis and Molecular Research Laboratory, Agricultural Sciences and Production Department, Zamorano University, San Antonio de Oriente, Francisco Morazán, Tegucigalpa 11101, Honduras; 3Centro de Estudios Biotecnológicos, Facultad de Ciencias Agropecuarias, Universidad de Matanzas, Autopista Varadero km 3 ½, Matanzas 44740, Cuba; 4National Center for Laboratory Animal Production, Santiago de las Vegas, Rancho Boyeros, La Habana 10100, Cuba; 5Poultry Research and Teaching Center, Agricultural Science and Production Department, Zamorano University, San Antonio de Oriente, Francisco Morazán, Tegucigalpa 11101, Honduras

**Keywords:** Creole bird, lactic acid bacteria, probiotic potential, antimicrobial activity, antimicrobial susceptibility

## Abstract

**Simple Summary:**

Although many countries have eliminated the use of antibiotics as growth promoters in poultry diets, more than 40 countries still use subtherapeutic antibiotics daily to reduce bacterial diseases triggered by the intensive production process, which has provoked microbial resistance and cross-resistance to other microorganisms. Thus, researchers continue to search for viable and feasible alternatives that also benefit the animals. Unquestionably, probiotics continue to be the main natural alternative. However, in many cases, the industry has rejected the use of these beneficial microorganisms due to the slight viability of bacterial strains and inconclusive results under different production conditions; thus, the use of native lactic acid bacteria from Creole birds, which have never been used as preventive antibiotics, may open up new tools for their widespread use in broiler production. This study showed that the isolation of *Lactobacillus reuteri* from the caeca of Creole roosters has high viability under different conditions of pH, bile salts, NaCl, and temperature, and in addition, this isolated bacterial strain strongly reduces the growth of pathogenic bacteria in vitro and has lower sensitivity to the use of three commonly used antibiotics, which allows their subsequent in vivo study in broilers.

**Abstract:**

Five strains (CLP2, CLP3, CLP4, CLP5, and CLP6) were isolated from the cecal content of Creole roosters fed without antibiotic growth promoters. Biochemical and morphological tests (negative catalase and oxidase) confirmed the presence of lactic acid bacteria. Additionally, considering the 16s RNA, *Lactobacillus vaginalis* (CLP2, CLP3, CLP5, and CLP6) and *Lactobacillus reuteri* (CLP4) were identified. All strains (mainly CLP4 and CLP5) showed variable and significant growth (*p* < 0.001) at different levels of pH. Likewise, all bacterial cultures were quantified at 42 °C, although only strains CLP4 and CLP5 managed to grow at 30 °C. Additionally, the CLP4, CLP5, and CLP6 strains grew from 0.05 to 0.30% of biliary salts. However, only the CLP4 isolate grew at different concentrations of NaCl (2–10%), and CLP5 grew at 2% NaCl. The CLP4 strain was able to inhibit the in vitro growth of enterobacteria such as *Escherichia coli* ATCC^®^ 11775TM, *Salmonella* Typhimurium ATCC^®^ 14028TM, and *Clostridium perfringens* ATCC^®^ 13124TM. In addition, CLP4 had lower sensitivity in the presence of amoxicillin and tetracycline compared to these pathogenic bacteria. Considering these in vitro results, it is necessary to carry out in vivo studies with the CLP4 strain to test the hypothesis of its probiotic effect in poultry.

## 1. Introduction

The growing demand for high-biological-value protein foods has caused the poultry industry to increase its production and look for efficient ways to improve performance without affecting animal health [1,2]. However, the growing gastrointestinal infections caused by pathogenic bacteria have led to an increase in the indiscriminate use of antibiotic growth promoters (AGPs), with the aim of preventing the development of diseases, reducing mortality, and maximizing the genetic potential of poultry (mainly in developing countries). Nevertheless, this has provoked greater bacterial resistance by transferring genes by genetic mutation to other pathogenic bacteria capable of colonizing the intestine and modifying the intestinal microbiota [3,4]. Moreover, these bacteria can be transmitted to humans, provoking zoonotic diseases such as salmonellosis [5]. Therefore, natural alternatives such as phytobiotics, organic acids, prebiotics, probiotics, and symbiotics are considered viable to eliminate or reduce the use of antibiotic growth promoters (AGPs) in diets, as well as to naturally prevent some gastrointestinal diseases [6,7].

Several studies indicate that lactic acid bacteria (LABs) can be safe alternatives to AGPs due to their multiple benefits to intestinal health by regulating digestive and metabolic processes and modulating the immune response and the production of proinflammatory cytokines, with improvements in the integrity of the intestinal mucosa by competitive exclusion [8,9,10]. 

To select a bacterial strain as a probiotic candidate, it is necessary to know the selection criteria and the results of in vitro tests that simulate the different conditions of the entire gastrointestinal tract (GIT) [11,12,13]. Thus, the bacterial strains must have certain characteristics that provide them with viability and survival in the gastrointestinal tract, among which are the ability to grow at different levels of pH and concentrations of bile salts [6]. Additionally, evaluations are carried out with different salinity concentrations of sodium chloride (NaCl) and at temperatures below and above 37 °C to ensure their viability in the environment and/or industrial processes [14]. Likewise, it is necessary to validate that LABs have antagonistic capacity against pathogenic bacteria as well as to conduct antimicrobial susceptibility tests to determine their interaction with antibiotics [15,16], which would justify their adhesion and colonization capacity in the intestinal walls. Furthermore, the presence of resistance genes should be investigated to make sure that they are not carriers of resistance themselves.

In this sense, Rajoka et al. [17] found high survival of some of the isolated LABs at different pH conditions, which resembled the gastrointestinal tract (GIT) of poultry and contributed to microbial eubiosis in in vivo tests. Furthermore, Ruben et al. [5] reported that isolated bacterial strains showed high survival in in vitro tests at different concentrations of gastric juice and bile salts, and they also increased adhesion to epithelial cells and stimulated autoaggregation and coaggregation. Moreover, Feng et al. [18] indicated that three strains of lactic acid bacteria reduced the adhesion and invasion of *Salmonella* spp. in vitro due to the antimicrobial effect of bacteriocins.

On the other hand, few studies have considered the isolation of bacterial strains as probiotic candidates from the gastrointestinal tract (mainly from the ceca) of Creole animals. These animals are fed a diverse diet (rich in prebiotics) without subtherapeutic antibiotics, which should favor the richness and diversity of the intestinal microbiota and its antimicrobial properties against *Enterobacteriaceae*, as well as reduce the risk of finding a potential strain with some resistance genes [19]. Therefore, the objective of this study was to evaluate the in vitro probiotic potentialities of autochthonous bacteria from the cecal content of Creole roosters for the substitution of antibiotics as a subtherapeutic alternative in poultry farming.

## 2. Materials and Methods 

### 2.1. Sample Collection and Preparation 

Five apparently healthy Creole roosters with an average age of 15 months and an average weight of 1.85 ± 0.5 kg were selected from the periurban areas of Zamorano, Francisco Morazán, Honduras, and raised in a natural environment free of AGPs.

Cervically stunning roosters fasted for 12 h were sacrificed, although water was supplied ad libitum. Then, a necropsy was performed to remove both ceca, which were placed in sterile bags and stored in a cold compartment. Samples (ceca) were transferred to the laboratory for in vitro testing. Under a sterile environment, the cecal content of the ceca was extracted, the pH was measured, and the entire content was placed in an Erlenmeyer flask with 30 mL of 1% (p/v) peptone water (Acumedia) shaking on the orbital table for 20 min at 150 rpm.

### 2.2. Isolation, Morphology, and Biochemical Tests

Serial dilutions in peptone water (Acumedia) to a concentration of 10^7^ from the stock solution were made. From each dilution, 100 µL of the solution was spread on Man, Rogosa, and Sharpe [20] culture medium (MRS Liufilchem) in duplicate, and plates were incubated at 37 °C for 48 h in a BD GasPakTM EZ anaerobic chamber.

Subsequently, preliminary biochemical tests of oxidase and catalase [21] were carried out, and the colonies that showed a negative reaction were Gram-stained and observed under light microscopy to confirm the morphological characteristics and the coloration of the lactic acid bacteria.

### 2.3. Genomic Identification 

Individual reactions from the duplicate isolates for their amplification were carried out, using components and conditions established by the Phusion Master Mix (M0531S) (Thermo Scientific™, Waltham, MA, USA) protocol. Sequences from the 16S ribosomal region were made using Thermocycler AB 2720 (Applied Biosystems™, Waltham, CA, USA). The 1465 bp region of the ribosomal gene 16S was amplified by polymerase chain reaction (PCR) using universal primers F27 (50-AGAGTTTGAT CMTGGCTCAG-30) and R1492 (50-TACGGYTACCTTGTTACGACTT-30), and its fragments were purified. The clean product of each sample was used with a modified cycle sequencing protocol for Big Dye Terminator v3.1. 

For sequencing, the PCR product was cleaned with Exo I and rSAP enzymes for each sample according to the manufacturer’s instructions. Then, to precipitate the cycle sequencing product, the described protocol was followed using EDTA at 125 mM. The pellet was resuspended in Hi-Di™ Formamide and read by SeqStudio ABI 3200 sequencer (Applied Biosystems, Waltham, CA, USA), using the Sanger sequencing methodology of capillary electrophoresis.

### 2.4. Tolerance to Different Growing Conditions

To evaluate the in vitro tolerance (to temperature, pH, sodium chloride, and bile salt) of the isolated strains, they were cultured on an MRS medium. The isolated strains were inoculated in test tubes containing 10 mL of MRS broth and incubated for 24 h under anaerobic conditions. For pH tolerance, concentrations of 2, 3, 4, 5.6, 6, and 7 were adjusted, using 0.1 N hydrochloric acid (HCl) and 0.1 N sodium hydroxide (NaOH). The tolerance concentrations of sodium chloride (NaCl) were 2, 4, 7, and 10% p/v. The temperatures evaluated were between 30 and 42 °C. For the evaluation of bile salts (catalog no. B8756-50G, Sigma Aldrich, St. Louis, MO, USA), a pH of 5.6 was adjusted to concentrations of 0.05, 0.1, 0.15, and 0.30% p/v. The strains were inoculated in triplicate, and serial dilutions of 10^4^ to 10^7^ of each sample were made and cultured in Petri dishes with MRS agar (Acumedia). Survival was determined by quantifying the colonies formed (CFU/mL).

### 2.5. Antimicrobial Activity Test 

#### 2.5.1. Activation of Strains

The bacterial strain (CLP4) that grew best in the in vitro tests was taken to perform the antagonism test against bacteria such as *E. coli* ATCC^®^ 11775TM, *Salmonella Typhimurium* ATCC^®^ 14028TM, and *Clostridium perfringens* ATCC^®^ 13124TM. To activate the pathogenic strains, a pellet was taken and dissolved in 0.5 mL of peptone water (Acumedia), and then a swab was saturated with the hydrated content and inoculated on plates with a selective culture medium for each strain of Hecktoen (*Salmonella*), McConkey (*E. coli*), and Clostridium (*Clostridium*). 

The plates were incubated aerobically at 37 °C for 24 h (LYFO DISCTM, Microbiologist), the most representative colonies of each strain were taken according to a loop and shaken in TSB broth, and then they were incubated aerobically for 18 h at 37 °C. The probiotic strain (CLP4) was activated by taking fresh colonies and inoculating them in TSB broth aerobically for 18 h at 37 °C [22].

#### 2.5.2. Antimicrobial Effect of the Probiotic Strain against Pathogenic Strains

After the activation of CLP4 and the pathogenic strains, the inhibition assay was prepared, 10 µL of the probiotic strain was taken and placed in the center of the Petri dish with MRS agar culture medium, and it was left to dry in the chamber of laminar flow and incubated at 37 °C in aerobiosis for 18 h. Then, 1 mL of each pathogenic strain was inoculated in duplicate in tubes with 9 mL of semisolid TSB agar at 37 °C, and the content of the tubes was poured into the Petri dishes with the probiotic strain, forming a double layer; subsequently, they were incubated under aerobic conditions for 18 h. The antagonistic activity between the probiotic strain and the pathogenic ones was verified by the formation of the inhibition halo, expressed in millimeters [22].

### 2.6. Antimicrobial Susceptibility Test 

To perform the antimicrobial susceptibility test, the Kirby–Bauer agar disc diffusion method [23] was used. The isolated bacterial strain and the pathogenic strains were subjected to different antibiotics commonly used to treat enteric and respiratory infections, including 30 µg of amoxicillin, 30 µg of tetracycline, and 30 µg of ampicillin. The probiotic strain and the pathogenic ones were sown massively in Mueller–Hinton agar. Then, disks impregnated with each antibiotic were distributed on the surface of the cultures, refrigerated for 30 min at 15 °C, and incubated at 40 °C for 24 h under aerobic conditions. After the incubation, the reading was made by measuring the diameter of the zone of inhibition expressed in millimeters.

### 2.7. Statistical Analysis 

The study is considered a completely randomized design. To determine the data normality and the variance uniformity, the Kolmogorov–Smirnov and Bartlett tests were used. Next, the data were processed by a simple classification analysis of variance (ANOVA). Where necessary, a post hoc analysis (Duncan) was used. All analyses were performed according to the statistical software IBM SPSS Statistics version 23.0.

## 3. Results

### 3.1. Isolation, Biochemical Test, and Genomic Identification 

Of the strains initially isolated, only six strains on MRS agar plates were seen as small, round, white, and creamy colonies. Additionally, these strains were Gram-positive, reacted negatively to catalase and oxidase, and had a rod-shaped morphology in short or single chains. Likewise, the sequencing of the bacterial strains identified five *Lactobacillus vaginalis* and one *Lactobacillus reuteri* with a homology of 97.36% (Table 1), which was registered on the GenBank website with the accession number: OQ134763.

### 3.2. Evaluation of the Probiotic Characteristics of Bacterial Strains 

Table 2 shows the growth of the isolated strains, where four survived the different pH conditions (CLP4, CLP5, and CLP6), with the best results for CLP4. Additionally, only the CLP4 and CLP5 strains grew under temperatures of 30 °C and 42 °C, with the highest quantifications for the first bacterial culture. Furthermore, this strain (CLP4) showed the best growth at different concentrations of NaCl and bile salts compared to the other isolates. Considering the results, the CLP4 strain was taken for the susceptibility and antagonism tests.

### 3.3. Antagonism Test 

Table 3 indicates that the selected strain with probiotic characteristics (CLP4) showed antagonistic activity against *Clostridium perfringens* and *Enterobacteriaceae*, with inhibition halos of 11.5, 12.5, and 14.00 mm against *Escherichia coli* ATCC^®^ 11775TM, *Salmonella* Typhimurium ATCC^®^ 14028TM, and *Clostridium perfringens* ATCC^®^ 13124TM, respectively, and without notable statistical differences (*p* > 0.05). 

### 3.4. Evaluation of the Antibiotic Susceptibility of the Probiotic Candidate

Selected strains were analyzed for their susceptibility to three antibiotics commonly used in the poultry industry (Table 4). All bacterial strains were susceptible to antibiotics such as amoxicillin, ampicillin, and tetracycline at a concentration of 30 µg. Additionally, the *Clostridium perfringens* ATCC 13,124 strain had the highest susceptibility to amoxicillin compared to the other bacterial strains. Likewise, the use of antibiotics such as ampicillin and tetracycline showed the greatest inhibition halo for the growth in plates of *Salmonella* Typhimurium ATCC 14028. It is important to note that the strain of *Lactobacillus reuteri* (CLP4) isolated from the caeca of Creole roosters had low susceptibility compared to pathogenic bacteria against three antibiotics.

## 4. Discussion

The aim of the study was to isolate bacterial strains with possible probiotic characteristics from the cecal content of Creole roosters that have not consumed growth-promoting antibiotics to be used later in broilers. The proper selection of probiotic strains has a direct effect on the intestinal health of poultry, either through an antimicrobial or anti-inflammatory effect [23]. In this sense, LABs are known for their efficiency and safety as probiotics in animals; however, in vitro evaluation is necessary because their properties and mechanisms of action in the gastrointestinal tract depend on the strain used [23,24]. Therefore, the probiotic properties of several bacterial isolates were investigated, and out of a total of nine isolated and cultivated strains, five showed the highest viability in growth at the beginning of the biochemical tests (Table 2). 

In the present study, it was found that the isolated strains corresponded to the genus *Lactobacillus* due to the negative results of the catalase and oxidase tests and the positive results of the Gram stain as well as the morphology of the colonies [10]. Likewise, 16S rRNA sequencing confirmed the *Lactobacillus* genus, which identified four *Lactobacillus vaginalis* strains and one *Latobacillus reuteri*. Specifically, it is described that the largest number of probiotics in the poultry industry comes from the *Lactobacillaceae* family; due to their multiple benefits for the health and productivity of the host animal, these bacterial strains can colonize the gastrointestinal tract and increase competitive exclusion, with immunomodulatory, anti-inflammatory, and antioxidant activities, which cause a natural effect of growth promotion [23,25]. 

Tolerance to different concentrations of pH and bile salts is one of the decisive criteria for selecting probiotic strains, especially to predict the probability of surviving adverse conditions in the GIT [26,27]. The pH in the GIT of poultry ranges between 1.8 and 7.0; thus, the bacterial strains that are candidates for probiotics must be able to survive during their journey through the digestive tract [23,28]. It is known that the viability of strains to changes in pH is highly variable and depends on the characteristics of the isolated bacterial colonies [27]. In this sense, of the six isolates, only five showed growths at pH levels between 2 and 7 (Table 2). The *Lactobacillus reuteri* strain (CLP4) showed the highest viability at different concentrations compared to the other isolates. Generally, lactic acid bacteria, due to their acidophilic condition, tolerate low pH concentrations as well as a high concentration of free acid (H+), which inhibits growth [28]. García-Hernandez [10] and Neethu et al. [29] found that the bacterial strains identified as *Lactobacillus pentosus*, *Lactobacillus reuteri*, *Lactobacillus rhamnosus*, and *Lactobacillus plantarum* survived at pH concentrations between 2 and 3; this is because lactic acid bacteria (mainly *Lactobacillus* spp.) produce organic acids such as acetic, butyric, propionic acids and lactic acid that reduce the pH by the emission of protons, which contributes to the tolerance of bacterial strains to different pH levels [28,30,31]. 

On the other hand, three of the isolated strains showed good tolerance to different concentrations of bile salts; however, the growth capacity decreased with increasing salt concentration (0.05–0.30%; Table 2). Bacterial growth at an exposure of 0.30% bile salts translates into high survival in the GIT and good probiotic potential of the candidate strains [10]. It is known that the concentration of bile salts in the poultry gut ranges between 0.2% and 0.3% [7]; thus, the isolated bacterial cultures (mainly *Lactobacillus reuteri* strain CLP4) showed significant growth (*p* < 0.001) up to a concentration of 0.30% of bile salts. The survival of bacterial strains at high concentrations of bile salts favors the ability to colonize the intestinal wall of the host as well as improves metabolic capacity because probiotics can participate in the hydrolyzation of conjugated bile salts and, in turn, the digestion of lipids [32,33]. Similar results were found by Reuben et al. [5], who observed significant growth at a high concentration of bile salts; on the contrary, studies by Dowarah et al. [34] did not find tolerance of the bacterial strains isolated up to a concentration of 0.30% of bile salts, even though they had survived at low pH concentrations. Betancur et al. [19] recommended isolating bacterial strains from the large intestine (mainly from the cecum) because the viable lactic acid bacteria in this intestinal portion have survived different conditions of pH and concentrations of bile salts, which could guarantee their viability in in vivo studies.

Similar to the other in vitro tests, the CLP4 strain showed the best results when a concentration between 2 and 10% NaCl was used. Reuben et al. [5] found no bacterial growth in vitro when using up to 10% NaCl, with the best results corresponding to a concentration of 6.5%. Similarly, Gandhi et al. [35] concluded that high concentrations of NaCl significantly decrease the viability of *Lactobacillus* spp., and they indicated that the maximum tolerable concentration was 3.5% NaCl. On the other hand, Betancur et al. [19] obtained high viability of bacterial culture isolated from the cecum of a Creole pig when they used up to a concentration of 10% NaCl, and they suggested that these strains can be used as preservatives for the meat, vegetables, and dairy industry and as probiotics used directly in drinking water or feed [36]. In this sense, our results indicated that the bacterium *Lactobacillus reuteri* isolated from the caeca of a Creole rooster has high viability at a high concentration of NaCl, which means that it could be used for industrial purposes and survive in more hostile environments [11]. However, other studies are needed to confirm this hypothesis.

Likewise, the viability of the CLP4 and CLP5 strains indicated the ability of these probiotic candidates to grow at hostile temperatures (30 and 42 °C), while the other bacterial strains had variable viability (Table 2). To achieve colonization of lactic acid bacteria in the GIT and improve the intestinal health of the animal host, they must be stable and proliferate in the intestine at temperatures above 37 °C [19]. In addition, the increase in temperatures due to microbial fermentation could decrease the contamination of pathogenic microorganisms, which is beneficial for the animal organism [37]. Furthermore, these results suggest that *Lactobacillus reuteri* CLP4 (mainly) could maintain its viability at different temperatures considering various vehicles and technological procedures where high temperatures are applied, with greater emphasis on drinking water, because elevated pelleting temperatures could decrease the viability of bacterial strains [38]. Reuben et al. [5] indicated that all isolated bacterial cultures grew at an optimal temperature of 37 °C after 24 h of incubation; however, under other temperature conditions, bacterial viability decreased significantly. Likewise, García-Hernández [11] found that *Lactobacillus pentosus* strain LB-31 grew at high temperatures, and the authors justified that proliferation under these conditions could benefit the growth of LABs in the GIT and the production of metabolites (bacteroids) in in vivo tests (broilers).

On the other hand, infections by Gram-positive pathogens such as *Clostridium* and Gram-negative pathogens such as *E. coli* and *Salmonella* spp. can provoke high mortality and significant economic losses in the poultry industry; thus, one of the main requirements for the selection of a probiotic strain is its ability to inhibit the growth of pathogenic bacteria [39], and in vitro studies can predict its possible antimicrobial effect on the host microbiota. Our results showed that the strain *L. reuteri* CLP4 showed antagonistic activity (Table 3), inhibiting common pathogenic strains in the poultry industry such as *E. coli* (ATCC 11775), *Salmonella* Typhimurium (ATCC 14028), and *Clostridium perfringens* (ATCC 13124). Other studies [10,32,40] have indicated antagonistic activity by strains of *Lactobacillus* spp. against different pathogens in broilers. However, some studies showed that LABs had antagonistic activity against *Clostridium* but not against *Salmonella* spp. and *E. coli* [36]. This antagonistic capacity of LABs is due to the production of different acids by the action of the fermentation process, such as lactic acid, short-chain fatty acids, and propionic and acetic acids, which prevents or reduces the growth of pathogenic bacteria in the gut. In addition, bacteriocins have antimicrobial properties by producing peptides synthesized in the ribosome of probiotic bacteria, with specific mechanisms to inhibit pathogenic strains [41,42]. Likewise, Betancur et al. [19] reported that *Lactobacillus plantarum* strains isolated from Creole animals decreased the growth of *E. coli* strain NBRC 102203, *S. enterica serovar* Typhimurium 4.5.12, *Klebsiella pneumoniae* ATCC BAA-1705D-5, and *Pseudomonas aeruginosa* ATCC 15442 due to the production of primary metabolites such as lactic acid, CO_2_ C_2_H_5_OH, and other bactericidal compounds.

Similarly, the poultry industry is interested in knowing the interaction of probiotic bacterial strains with subtherapeutic and therapeutic antibiotics commonly used in diets and drinking water [8]. Additionally, there is high intrinsic resistance and susceptibility of the different probiotic microorganisms; thus, it is important to know the activity against various antibiotics, mainly penicillin derivatives [43], with the aim of elucidating which antibiotics could be used to maintain the survival capacity of the probiotics to different environments [8,44]. In this study, it was shown that the probiotic candidate *Lactobacillus reuteri* (CL4) had a lower susceptibility than the pathogenic strains of *E. coli* ATCC 11775 and *Salmonella* ATCC 14028 (Table 4) to the antibiotics such as amoxicillin and tetracycline and also had a similar response to *Clostridium* ATCC 13,124 when ampicillin was used. Dowarah et al. [34] reported high susceptibility to penicillin and ampicillin in *Lactobacillus* spp. isolated from pigs and broilers. Furthermore, 78% of bacterial isolates have been reported to be resistant to tetracycline such as *L. reuteri*, *L. gallinarum*, *L. crispatus*, and *L. salivarius* [45,46]. Likewise, Betancur et al. [19] mentioned that strains isolated from Creole animals showed susceptibility to amoxicillin (24.3–26.7 mm); otherwise, when ciprofloxacin and tetracycline were used, the strains were resistant (6.8–8.1 mm). Considering that many countries currently use daily subtherapeutic antibiotics in apparently healthy animals (mainly the US and Latin America), these authors recommend the use of viable probiotic bacterial strains against these antibiotics (growth promoters or therapeutic antibiotics) [6,7]. Moreover, probiotic studies against antibiotics allow us to know their interaction in the host animal, since many antibiotics are increasingly ineffective in the control of diseases caused by some bacterial pathogens. Furthermore, the use of high concentrations of antibiotics provokes bacterial resistance when transferring resistance between bacteria of different genes through conjugated plasmids or chromosomes [47,48]; thus, probiotic strains are an alternative by showing efficiency and safety for animals [49].

## 5. Conclusions

Of the strains isolated from the cecal content of Creole roosters, only the *Lactobacillus reuteri* strain (CLP4) identified by biochemical and molecular tests indicated good growth under different conditions of pH, bile salts, NaCl, and temperature. Additionally, this bacterial culture (CLP4) inhibited the growth of pathogenic bacteria in vitro as well as showed lower susceptibility compared to pathogenic bacteria against several antibiotics commonly used in the poultry industry. Based on these findings, a study of the effect of the microbial isolate (CLP4) in the drinking water of broilers was carried out to verify the probiotic effects under production conditions.

## Figures and Tables

**Table 1 animals-13-00455-t001:** Sequencing of bacterial isolates by 16s rRNA from cecal content in Creole rooster.

Organisms	Score	Similarity	Value E	Coincidences
	NCBI	NCBI	NCBI	NCBI
*Lactobacillus vaginalis* (CLP2)	260	91.67%	5e-65	44
*Lactobacillus vaginalis* (CLP3)	1104	95.67%	0	44
*Lactobacillus reuteri* (CLP4)	1086	97.36%	0	95
*Lactobacillus vaginalis* (CLP5)	737	97.24%	0	58
*Lactobacillus vaginalis* (CLP6)	1077	96.51%	0	51

**Table 2 animals-13-00455-t002:** In vitro assessment of functional properties of *Lactobacillus* spp. isolates from the cecal content of Creole roosters under various pH, bile salt, sodium chloride (NaCl), and temperature conditions.

Items	Growth (Log CFU/mL) of Isolated Bacterial Strains					SEM±	*p* Value
	CLP 2	CLP 3	CLP4	CLP5	CLP6		
pH							
2	7.67 ^b^	7.54 ^c^	7.87 ^a^	6.65 ^d^	7.57 ^c^	0.10	<0.001
3	10.21 ^a^	8.91 ^d^	9.58 ^b^	9.65 ^b^	9.40 ^c^	0.02	<0.001
4	^____^	10.2 ^a^	10.2 ^a^	8.91 ^c^	9.94 ^b^	0.01	<0.001
5.6	^____^	9.94 ^ab^	10.4 ^a^	10.3 ^b^	10.2 ^c^	0.02	<0.001
6	9.31 ^d^	7.98	10.8 ^a^	10.2 ^b^	10.1 ^c^	0.02	<0.001
7	8.56 ^b^	7.43 ^d^	8.79 ^a^	8.54 ^b^	7.70 ^c^	0.02	<0.001
Temperature (°C)							
30 °C	^____^	^_____^	10.4 ^a^	7.23 ^b^	^_____^	0.02	<0.001
42 °C	7.63 ^c^	7.24 ^d^	11.0 ^a^	7.99 ^b^	5.89 ^d^	0.05	<0.001
NaCl (%)							
2	^____^	^_____^	8.81 ^a^	5.75 ^b^	^_____^	0.08	<0.001
4	^____^	^_____^	9.87	^_____^	^_____^	0.05	>0.999
7	^_____^	^_____^	9.82	^_____^	^_____^	0.05	>0.999
10	^_____^	^_____^	9.17	^_____^	^_____^	0.05	>0.999
Bile salts (%)							
0.05	^_____^	^_____^	11.2 ^a^	10.3 ^b^	9.88 ^c^	0.03	<0.001
0.10	^_____^	^_____^	10.7 ^a^	10.3 ^b^	9.82 ^c^	0.07	0.003
0.15	^_____^	^_____^	9.50 ^a^	9.35 ^a^	6.75 ^b^	0.08	<0.001
0.30	^_____^	^_____^	9.15 ^a^	8.14 ^b^	6.78 ^c^	0.04	<0.001

^a,b,c,d^ Means with different letters are significantly different at *p* < 0.05, *n* = 6.

**Table 3 animals-13-00455-t003:** In vitro screening of antibacterial activity of CLP4 isolates from the cecal content of Creole roosters against enteropathogenic bacteria (*n* = 6).

Pathogenic Strains	*Lactobacillus reuteri* (CLP4)
	Inhibition Halo (mm)
*Escherichia coli* ATCC^®^ 11775^TM^	12.50
*Salmonella* Typhimurium ATCC^®^ 14028^TM^	14.00
*Clostridium perfringens* ATCC^®^ 13124^TM^	11.50
SEM±	0.850
*p* value	0.193

**Table 4 animals-13-00455-t004:** Evaluation of the antibiotic sensitivity of the *Lactobacillus* spp. isolates from the cecal content of Creole roosters (*n* = 6).

Strains	Antibiotics (mm) *		
	Amoxicillin	Ampicillin	Tetracycline
*Lactobacillus reuteri* (CLP4)	32.00 ^b^	34.00 ^b^	31.00 ^b^
*Escherichia coli* ATCC 11775	33.00 ^b^	35.00 ^b^	35.67 ^ab^
*Salmonella* Typhimurium ATCC 14028	36.00 ^b^	39.00 ^a^	40.00 ^a^
*Clostridium perfringens* ATCC 13124	41.00 ^a^	32.00 ^b^	33.00 ^b^
SEM±	1.443	1.155	1.509
*p* value	0.009	0.015	0.015

* Inhibition halo (mm). ^a,b^ Means with different letters in the same row differ at *p* < 0.05. SEM: standard error of the mean

## Data Availability

The data presented in this study are available on request from the corresponding author.

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
