# Peer review of "Characterization of Autochthonous Strains from the Cecal Content of Creole Roosters for a Potential Use as Probiotics"

_animals, 2023, doi:10.3390/ani13030455_

Round 1

Reviewer 1 Report

The manuscript is interesting given that bacterial resistance is a public health problem. However, some points must be addressed before the potential publication of the manuscript.

L.32. Check that the word "in vitro" and "in vivo" are in italics in the manuscript.

L.58. ...countries). Nevertheless, this...

L.107. Check the number, please. Probably is 1.85 ± 0.5 kg.

L.141. Tubes were adjusted to pH 1, 2, ...7 with 0.1N HCl or 0.1N NaOH, not the concentrations. Why did not the authors use buffer solutions to reach the pH? What was the contact time in the medium and what was it based on?

L.144-146. Please write the paragraph since it is not understood. The concentration is W/V, not P/V.

L.217. What is the strain? Clostridium?

L.300-303. Mention the contact times of the bacterial inoculum with the different conditions evaluated. Do the times used to carry out the experiment agree with studies already published?

L.316-319. It is clear that the temperature is important for maintaining the viability of the bacteria, but technologically speaking (pelletizing) it is a low temperature, thinking that the lactobacilli can be included in the feed. Are there viability data at temperatures above 40 ºC?

L.325. It is a fact that antagonism studies are important, but what happens if two or more strains of lactobacilli are combined, could the antimicrobial effect be enhanced, that is, a synergistic effect is present? How would you evaluate it? Would it be important for these effects to occur? What advantages would these effects have?

L.364. Is it important to include the field of application of the probiotic strain, that is, how will it be administered (drinking water or feed)? in what species will it be applied?

Author Response

Dear reviewer, thank you very much for your comments to improve our manuscript.

The manuscript is interesting given that bacterial resistance is a public health problem. However, some points must be addressed before the potential publication of the manuscript.

Reviewer: L.32. Check that the word "in vitro" and "in vivo" are in italics in the manuscript.

Authors: Done

Reviewer: L.58. ...countries). Nevertheless, this...

Authors: Done

Reviewer: L.107. Check the number, please. Probably is 1.85 ± 0.5 kg.

Authors: Done

Reviewer: L.141. Tubes were adjusted to pH 1, 2, ...7 with 0.1N HCl or 0.1N NaOH, not the concentrations.

Authors: Done. It was corrected

Reviewer: Why did not the authors use buffer solutions to reach the pH? What was the contact time in the medium and what was it based on?

Authors: We follow what is established and published by Betancur et al. (2020) in Animals.

Betancur, C., Martínez, Y., Tellez-Isaias, G., Avellaneda, M. C., & Velázquez-Martí, B. (2020). In vitro characterization of indigenous probiotic strains isolated from Colombian creole pigs. Animals10(7), 1204.

Reviewer: L.144-146. Please write the paragraph since it is not understood. The concentration is W/V, not P/V.

Authors: Done. It was corrected

Reviewer: L.217. What is the strain? Clostridium?

Authors: Done. It was corrected

Reviewer: L.300-303. Mention the contact times of the bacterial inoculum with the different conditions evaluated. Do the times used to carry out the experiment agree with studies already published?

Authors: We follow what is established and published by Betancur et al. (2020).

Betancur, C., Martínez, Y., Tellez-Isaias, G., Avellaneda, M. C., & Velázquez-Martí, B. (2020). In vitro characterization of indigenous probiotic strains isolated from Colombian creole pigs. Animals10(7), 1204.

Reviewer: L.316-319. It is clear that the temperature is important for maintaining the viability of the bacteria, but technologically speaking (pelletizing) it is a low temperature, thinking that the lactobacilli can be included in the feed. Are there viability data at temperatures above 40 ºC?

Authors: The manuscript specified that the further use of this lactobacillus is to be used in drinking water, like most lactobacillus-derived probiotics, except those that are encapsulated, which was not the case. It should be noted that the in vivo study was carried out in broiler drinking water.

Reviewer: L.325. It is a fact that antagonism studies are important, but what happens if two or more strains of lactobacilli are combined, could the antimicrobial effect be enhanced, that is, a synergistic effect is present? How would you evaluate it? Would it be important for these effects to occur? What advantages would these effects have?

Authors: Authors: The research team asked the same questions; we are currently interpreting the results of other LAB premix studies. In the scientific literature, it is still not clear if it is a LAB premix or if the inclusion of a bacterial strain has the same positive effects in the animals, which will clearly depend on the viability of the bacterial strain, the colonization in the GIT, and the effect about animals. However, in this study, it was concluded that CLP4 is the bacterium that survived the different study conditions simulating the GIT of poultry.

Reviewer: L.364. Is it important to include the field of application of the probiotic strain, that is, how will it be administered (drinking water or feed)? in what species will it be applied?

Authors: Done. “Based on these findings, a study of the effect of the microbial isolate (CLP4) in the drinking water of broilers was carried out to verify the probiotic effects under production conditions”.

Reviewer 2 Report

With the negative impact of antibiotic abuse in animal production and development of  antibiotics prohibition activity in worldwide, it is urgent to search alternatives to antibiotics, it is gratifying that the development and utilization of probiotic products provides an important alternatives for antibiotics. The experiment isolated several bacterial strains from cecal content of roosters and systematically performed tests on identification, antimicrobial activity and antimicrobial susceptibility. The manuscript is structurally complete, and well written. Some questions and suggestions are as follows:

Line 107:  an average weight of 1.85±5 kg please recheck the standard deviation or standard error of body weight.

Line 185-187:To determine the data normality and the variance uniformity, the Kolmogorov–Smirnov and Bartlett tests were used. For ANOVA analysis, data normality and variance uniformity are assumed for each group and the sample size for each group is five, please provided detailed figures or tables to demonstrate the normality of the data. Besides, sample size of Table 3 and Table 4 should be labeled clearly. 

Line 209-212: In Table 2, multiple comparisons were performed among five isolated strains on different pH, bile salt, sodium chloride (NaCl), and temperature conditions. For each strain, multiple comparison among different PH levels or other factor levels could also be performed, which would provided a more specific reference for further research.

Author Response

Dear reviewer, thank you very much for your comments to improve our manuscript.

With the negative impact of antibiotic abuse in animal production and development of antibiotics prohibition activity in worldwide, it is urgent to search alternatives to antibiotics, it is gratifying that the development and utilization of probiotic products provides an important alternatives for antibiotics. The experiment isolated several bacterial strains from cecal content of roosters and systematically performed tests on identification, antimicrobial activity and antimicrobial susceptibility. The manuscript is structurally complete, and well written. Some questions and suggestions are as follows:

Reviewer: Line 107: “an average weight of 1.85±5 kg” please recheck the standard deviation or standard error of body weight.

Authors: Done.

Reviewer: Line 185-187:“To determine the data normality and the variance uniformity, the Kolmogorov–Smirnov and Bartlett tests were used”. For ANOVA analysis, data normality and variance uniformity are assumed for each group and the sample size for each group is five, please provided detailed figures or tables to demonstrate the normality of the data. Besides, sample size of Table 3 and Table 4 should be labeled clearly. 

Authors: Data normality and variance homogeneity tests are carried out to verify if the difference found between the treatments corresponds to the source of initial statistical variation. If the data is not normal and the variance is not uniform, a transformation of the data is performed and the multiple comparison of means a posteriori is done with the transformed mean, which was not the case. The team considers that it is not necessary to incorporate a table that demonstrates these data. The sample size in table 3 and 4 is observed in their title (n=6)

Reviewer: Line 209-212: In Table 2, multiple comparisons were performed among five isolated strains on different pH, bile salt, sodium chloride (NaCl), and temperature conditions. For each strain, multiple comparison among different pH levels or other factor levels could also be performed, which would provided a more specific reference for further research.

Authors: Initially, we considered comparing not only the strains, but an interaction with the different variants of pH, NaCl, bile salts and temperature, however, the other strains did not have favorable results, except for CLP4, which could confuse the interpretation of the data. Also other studies that we take as a reference carry out studies similar to ours, leaving the isolated bacterial strains as the central axis.

Betancur, C., Martínez, Y., Tellez-Isaias, G., Avellaneda, M. C., & Velázquez-Martí, B. (2020). In vitro characterization of indigenous probiotic strains isolated from Colombian creole pigs. Animals10(7), 1204.

Tellez-Isaias, G.; Vuong, C.N.; Graham, B.D.; Selby, C.M.; Graham, L.E.; Señas-Cuesta, R.; Barros, T.L.; Beer, L.C.; Coles, M.E.; Forga, A.J.; et al. Developing probiotics, prebiotics, and organic acids to control salmonella spp. in commercial turkeys at the University of Arkansas, USA. Ger. J. Vet. Res. 2021, 1, 7–12. DOI: 10.51585/gjvr.2021.3.0014

Yaneisy GH.; Tania PS.; Ramón B.; José L.; Balcázar JR.; Nicoli JMS.; Isolation, characterization and evaluation of probiotic lactic acid bacteria for potential use in animal production. Res Vet Sci. 2016; 108:125–32. DOI: 10.1016/j.rvsc.2016.08.009.

Prado-Rebolledo, O.F.; de Jesus Delgado-Machuca, J.; Macedo-Barragan, R.J.; Garcia- Márquez, L.J.; Morales-Barrera, J.E.; Latorre, J.D.; Hernandez-Velasco, X.; Tellez, G. Evaluation of a selected lactic acid bacteria-based probiotic on Salmonella enterica serovar Enteritidis colonization and intestinal permeability in broiler chickens. Avian Pathol. 2017, 46, 90–94. DOI: 10.1080/03079457.2016.1222808

Kers, J.G.; Velkers, F.C.; Fischer, E.A.J.; Hermes, G.D.A.; Stegeman, J.A.; Smidt, H. Host and environmental factors affecting the intestinal microbiota in Chickens. Front. Microbiol. 2018, 9, 235. DOI: doi.org/10.3389/fmicb.2018.00235

Reviewer 3 Report

The manuscript is about the limited characterization of autochthonous strains isolated from the ceca content of roosters in the hope it would work as a probiotic.

Title should rather be: Characterization of autochthonous strains from the cecal content of creole roosters for a potential use as probiotics.

L20 antibiotics (with and “s”); add “as” before growth.

L24 “animal organism”, do we refer to the animal itself or its microbes; restructure.

L46 “in vitro” and “in vivo” in italic.

L79 Presence of resistance genes should be investigated to make sure they are not carriers of resistance themselves.

L92 and reduce the risk of finding a potential strain with some resistance genes.

Section 2.1 not useful, remove.

L106 Details on how the health conditions were determined should be provided.

L108-L110 Poor English language; needs to be fixed entirely.

L117-118 Change for:  spread on deMan, Rogosa and Sharpe… and plates were incubated at…  remove "were incubated" at the end.

L121 Changed for: were Gram stained and observed under light microscopy

L132-135 redundancy of the word use (3x)

Commercial description should be: (model, company, city, country)

L139-L141 Poor English language; needs to be fixed entirely.

L141-L142 Poor English language; needs to be fixed entirely; concentrations? Spaces are missing to separate the numbers.

OK I stop here with the English language to focus on the science. This is beyond my duty as a reviewer.

L152 How was growth measured? Plate counts or absorbance. I am assuming cell counts with Table 2.

L158 Coma missing at the end.

L201 Remove coma after in.

Fig.1 is awful.

Table 2 should not be spread on two pages. Idem table 4.

L215 Add a coma (,) after 11.5.

L351 Lactobacillus in italic

L356-357 Then what is the risk that the probiotic becomes a carrier of resistance genes that could be transferred?

L369 Is the lower susceptibility to antibiotics caused by resistance genes? If yes, they can be transferred… This should be determined before going into production. The entire genome of CLP4 should be sequenced and analyzed for that purpose.

Pipeline analysis for the 16S sequences should be detailed.

16S sequences should have been deposited in a collection such as Genebank, NCBI and details provided before submission.

Author Response

Dear reviewer, thank you very much for your comments to improve our manuscript.

The manuscript is about the limited characterization of autochthonous strains isolated from the ceca content of roosters in the hope it would work as a probiotic.

Reviewer: Title should rather be: Characterization of autochthonous strains from the cecal content of creole roosters for a potential use as probiotics.

Authors: Done.

Reviewer: L20 antibiotics (with and “s”); add “as” before growth.

Authors: Done.

Reviewer: L24 “animal organism”, do we refer to the animal itself or its microbes; restructure.

Authors: Done.

Reviewer: L46 “in vitro” and “in vivo” in italic.

Authors: Done.

Reviewer: L79 Presence of resistance genes should be investigated to make sure they are not carriers of resistance themselves.

Authors: Done.

Reviewer: L92 and reduce the risk of finding a potential strain with some resistance genes.

Authors: Done.

Reviewer: Section 2.1 not useful, remove.

Authors: Done.

Reviewer: L106 Details on how the health conditions were determined should be provided.

Authors: Done.

Reviewer: L108-L110 Poor English language; needs to be fixed entirely.

Authors: Done.

Reviewer: L117-118 Change for:  spread on deMan, Rogosa and Sharpe… and plates were incubated at…  remove "were incubated" at the end.

Authors: Done.

Reviewer: L121 Changed for: were Gram stained and observed under light microscopy

Authors: Done.

Reviewer: L132-135 redundancy of the word use (3x)

Authors: Done.

Reviewer: Commercial description should be: (model, company, city, country)

Authors: Done.

Reviewer: L139-L141 Poor English language; needs to be fixed entirely.

L141-L142 Poor English language; needs to be fixed entirely; concentrations? Spaces are missing to separate the numbers. OK I stop here with the English language to focus on the science. This is beyond my duty as a reviewer.

Authors: Done. The English was reviewed by an English-speaking expert.

Reviewer: L152 How was growth measured? Plate counts or absorbance. I am assuming cell counts with Table 2.

Authors: Done.

Reviewer: L158 Coma missing at the end.

Authors: Done.

Reviewer: L201 Remove coma after in.

Authors: Done.

Reviewer: Fig.1 is awful.

Authors: Figure 1 was removed, we consider that table 1 shows the genetic sequencing

Reviewer: Table 2 should not be spread on two pages. Idem table 4.

Authors: Done.

Reviewer: L215 Add a coma (,) after 11.5.

Authors: Done.

Reviewer: L351 Lactobacillus in italic

Authors: Done.

Reviewer: L356-357 Then what is the risk that the probiotic becomes a carrier of resistance genes that could be transferred?

Authors: It is known that despite the fact that some countries such as the European Union have prohibited the use of sub-therapeutic antibiotics, countries such as the US and most Latin American countries use these antibiotics, in practice combined with probiotics, for which this study can elucidate what association or resistance this strain of bacteria may have against common antibiotics. The wording has been changed for better understanding.

Reviewer: L369 Is the lower susceptibility to antibiotics caused by resistance genes? If yes, they can be transferred… This should be determined before going into production. The entire genome of CLP4 should be sequenced and analyzed for that purpose.

Authors: We followed the model of several authors to find which bacterin strain showed the greatest beneficial effect in vitro simulating the GIT, in a later study the study will be carried out to determine the microbial resistance, either by plasmid or chromosome.

Betancur, C., Martínez, Y., Tellez-Isaias, G., Avellaneda, M. C., & Velázquez-Martí, B. (2020). In vitro characterization of indigenous probiotic strains isolated from Colombian creole pigs. Animals10(7), 1204.

Tellez-Isaias, G.; Vuong, C.N.; Graham, B.D.; Selby, C.M.; Graham, L.E.; Señas-Cuesta, R.; Barros, T.L.; Beer, L.C.; Coles, M.E.; Forga, A.J.; et al. Developing probiotics, prebiotics, and organic acids to control salmonella spp. in commercial turkeys at the University of Arkansas, USA. Ger. J. Vet. Res. 2021, 1, 7–12. DOI: 10.51585/gjvr.2021.3.0014

Yaneisy GH.; Tania PS.; Ramón B.; José L.; Balcázar JR.; Nicoli JMS.; Isolation, characterization and evaluation of probiotic lactic acid bacteria for potential use in animal production. Res Vet Sci. 2016; 108:125–32. DOI: 10.1016/j.rvsc.2016.08.009.

Prado-Rebolledo, O.F.; de Jesus Delgado-Machuca, J.; Macedo-Barragan, R.J.; Garcia- Márquez, L.J.; Morales-Barrera, J.E.; Latorre, J.D.; Hernandez-Velasco, X.; Tellez, G. Evaluation of a selected lactic acid bacteria-based probiotic on Salmonella enterica serovar Enteritidis colonization and intestinal permeability in broiler chickens. Avian Pathol. 2017, 46, 90–94. DOI: 10.1080/03079457.2016.1222808

Kers, J.G.; Velkers, F.C.; Fischer, E.A.J.; Hermes, G.D.A.; Stegeman, J.A.; Smidt, H. Host and environmental factors affecting the intestinal microbiota in Chickens. Front. Microbiol. 2018, 9, 235. DOI: doi.org/10.3389/fmicb.2018.00235

Reviewer: Pipeline analysis for the 16S sequences should be detailed.

Authors: Done

Reviewer: 16S sequences should have been deposited in a collection such as GenBank, NCBI and details provided before submission.

Authors: Thanks for your suggestion. We registered the CLP4 strain in GenBank with code OQ134763. https://submit.ncbi.nlm.nih.gov/subs/genbank/SUB12443853/overview